# Prolonged Inhibition of the MEK1/2-ERK Signaling Axis Primes Interleukin-1 Beta Expression through Histone 3 Lysine 9 Demethylation in Murine Macrophages

**DOI:** 10.3390/ijms241914428

**Published:** 2023-09-22

**Authors:** Rachel Low, Soon-Duck Ha, Nichita Sleapnicov, Parthiv Maneesh, Sung Ouk Kim

**Affiliations:** Department of Microbiology and Immunology, University of Western Ontario, London, ON N6G 2V4, Canada; rlow7@uwo.ca (R.L.); sha3@uwo.ca (S.-D.H.); nsleapni@uwo.ca (N.S.); pmaneesh@uwo.ca (P.M.)

**Keywords:** macrophages, LPS, MEK1/2, ERK, IL-1β, H3K9 methylation, CBX5, innate immune memory, tolerance, priming, training

## Abstract

Macrophages undergo different cellular states upon activation that can be hyporesponsive (tolerated) or hyperresponsive (primed or trained) to subsequent stimuli. Epigenetic modifications are known to play key roles in determining these cellular states. However, little is known about the role of signaling pathways that lead to these epigenetic modifications. Here, we examined the effects of various inhibitors targeting key signaling pathways induced by lipopolysaccharide (LPS) on tolerance and priming in murine macrophages. We found that a prolonged inhibition (>18 h) of the mitogen-activated protein kinase (MEK)1/2—extracellular signal-regulated kinase (ERK)1/2 signaling axis reversed tolerance and primed cells in expressing interleukin (IL)-1β and other inflammatory cytokines such as IL-6, tumor necrosis factor (TNF)α, and CXCL10. The ectopic expression of catalytically active and inactive MEK1 mutants suppressed and enhanced IL-1β expression, respectively. A transcriptomic analysis showed that cells primed by the MEK1/2 inhibitor U0126 expressed higher levels of gene sets associated with immune responses and cytokine/chemokine production, but expressed lower levels of genes with cell cycle progression, chromosome organization, and heterochromatin formation than non-primed cells. Of interest, the mRNA expressions of the histone 3 lysine 9 (H3K9) methyltransferase *Suv39h1* and the H3K9 methylation reader *Cbx5* were substantially suppressed, whereas the H3K9 demethylase *Kdm7a* was enhanced, suggesting a role of the MEK1/2-ERK signaling axis in H3K9 demethylation. The H3K9 trimethylation levels in the genomic regions of IL-1β, TNFα, and CXCL10 were decreased by U0126. Also, the H3K9 methyltransferase inhibitor BIX01294 mimicked the U0126 training effects and the overexpression of chromobox homolog (CBX)5 prevented the U0126 training effects in both RAW264.7 cells and bone-marrow-derived macrophages. Collectively, these data suggest that the prolonged inhibition of the MEK1/2-ERK signaling axis reverses tolerance and primed macrophages likely through decreasing the H3K9 methylation levels.

## 1. Introduction

Macrophages are sentinel innate immune cells that orchestrate immune responses in almost all tissues. Their diversity in function requires plasticity and resilience to adapt to changing microenvironments. A cellular state that renders rapid and robust inflammation is referred to as “trained” or “primed”, whereas an attenuated and anti-inflammatory cellular state is referred to as “tolerated”. These altered cellular states are crucial for fending off infections and maintaining immune homeostasis, respectively [1,2]. Upon exposure to microenvironmental and microbial stimuli, macrophages undergo intrinsic changes, rendering different cellular states. Several stimuli, including β-glucan, Bacillus Calmette–Guérin (BCG) vaccine, and oxidized low-density lipoprotein (oxLDL), render macrophages to a trained state that enhances inflammatory responses when cells are subsequently exposed to microbial components such as lipopolysaccharide (LPS) [3,4]. Conversely, tolerance is demonstrated when macrophages that were pre-exposed to LPS become less responsive to the same or similar stimuli in producing inflammatory cytokines [5]. The first level of tolerance is a general dampening of activation and signaling through transient (lasting hours) negative feedback mechanisms including the downregulation of receptors [6], expressing negative signaling regulators, such as A20 [7] and interleukin-1 receptor-associated kinase-M (IRAK-M) [8], and inducing repressive transcription factors, such as B cell leukemia-3 and nuclear factor-κB (NF-κB) p50 [9]. The second level of tolerance is manifested at a later stage, where tolerance stimuli render prolonged (lasting days or weeks) and gene-specific changes that repress or enhance transcription through epigenetic mechanisms [10,11,12].

To date, extensive studies have revealed various epigenetic modifications, such as histone modifications, DNA methylation, and non-coding RNA expression, that determine macrophage states [10,13]. Among them, histone modifications, particularly in H3K4 mono- or tri-methylation (H3K4me1/3) and H3K27 acetylation (H3K27ac), are transactivating markers that are involved in training/priming, whereas H3K9 di/tri-methylations (H3K9me2/3) are repressive markers of tolerated genes [14,15,16]. In the human monocytic THP-1 cell line, prolonged LPS stimulation suppresses the expressions of TNFα [17,18] and IL-1β [19] by inducing H3K9 methylation through recruiting the histone H3K9 methyltransferase euchromatic histone lysine N-methyltransferase 2 (EHMT2, also known as G9a) and heterochromatin binding protein 1 α (HP1α, also known as chromobox (CBX)5) in their promoters and enhancers. In contrast, the dissociation of G9a in the promoters/enhancers is required for expressing late-response genes induced by LPS [20]. In murine macrophages, however, tolerance is mainly mediated by the loss of the transactivating markers H3K4me1/2 [12] and H3K27ac [4], rather than the gain of the repressive H3K9me2/3 markers [12]. 

In mammals, LPS initiates intracellular signaling cascades by binding to its receptor, Toll-like receptor 4 (TLR4) [21]. Upon activation, TLR4 triggers multiple signaling cascades through the myeloid differentiation primary response protein 88 (MyD88) and toll/IL-1 receptor domain-containing adapter inducing interferon (TRIF) adaptors [22,23]. The key downstream signaling axes include the inhibitor kB kinases (IκK)-NF-κB axis, phosphatidylinositol 3-kinase (PI3K)-AKT axis, the three mitogen-activated protein kinase (MAPK) signaling axes (MAPK kinase (MEK)1/2-extracellular signal-regulated protein kinases (ERK), MEK3/6-p38, and MEK4/7-cJun N-terminal kinase (JNK)), and the antiviral TRIF-IRF3 signaling axis [24,25,26]. The activation of these signaling axes is crucial for anti-microbial and proinflammatory responses by inducing effector molecules at various levels including transcription factors, transactivation machinery formation, translation, RNA stability, and protein function [27,28,29]. In general, the IκK-NF-κB and MEK3/6-p38 axes are crucial for proinflammatory and anti-infective responses; the MEK4/7-JNK axis is for cell death and proinflammation; and the MEK1/2-ERK axis is for cell proliferation and proinflammation [30]. The PI3K-protein kinase D-AKT pathway contributes both proinflammatory and regulatory responses depending on the isoforms involved in the pathway: AKT1 inhibits LPS responses and promotes immune-regulatory effector functions; AKT2 has opposing roles of AKT1 [31]. To date, extensive studies have shed light on the role of these signaling axes in inflammation and gene transcription. However, how these signaling axes contribute to macrophage tolerance and priming/training is yet to be defined.

This study examined the role of key TLR4 signaling axes in LPS-induced tolerance and training of macrophages using IL-1β expression as a readout. IL-1β (derived from the proteolytic processing of pro-IL-1β gene product; herein also referred to as IL-1β) is a potent inflammatory cytokine and is rapidly induced by TLR4 activation. We found that the inhibition of the MEK1/2-ERK signaling axis had dual effects on the expression of IL-1β in murine macrophages: an early/short-term inhibition of the signaling axis suppressed IL-1β expression but prolonged inhibition (>18 h) primed IL-1β expression. The priming effects were mediated by decreasing the H3K9me3 levels in part through downregulating CBX5 expression.

## 2. Results

### 2.1. Inhibition of MEK1/2 Enhances IL-1β Expression in LPS-Tolerized RAW264.7 Macrophages

To examine the role of key signaling cascades involved in LPS-induced tolerance, we first induced tolerance by treating RAW264.7 macrophages with a low dose of LPS (10 ng/mL) for 18–24 h. After removing LPS, these cells were replated and further cultured with fresh cell media for 4–6, 24, 48, and 72 h. These cells were then activated with a high dose of LPS (100 ng/mL) for 4–6 h (Figure 1A, upper panel), and the expression of IL-1β mRNA was examined via qPCR (Figure 1A, lower panel). Compared to the non-tolerized cells, the tolerized cells were significantly compromised in expressing IL-1β mRNA for up to 48 h. We then examined the role of TLR4 signaling pathways by inducing tolerance in the presence of inhibitors targeting key signaling pathways (Figure 1B). Among them, the MEK1/2 inhibitors, U0126 and selumetinib, were able to inhibit tolerance. Since the activation of the MEK1/2 signaling is the most prominent at the early phase of LPS stimulation, we examined if early inhibition (exposing U0126 in the first 6 h of tolerance) or late inhibition (exposing U0126 in the last 6 h of tolerance) was enough to inhibit tolerance (Figure 1C, upper panel). However, the tolerance-reversing effect was only observed when U0126 was treated for a prolonged time (18–24 h) (Figure 1C, lower panel). As expected, the MEK1/2, JNKs, and p38 inhibitors significantly suppressed IL-1β mRNA expression when the cells were exposed during the 6 h activation phase (Figure 1D). The activation of ERK and NF-κB by LPS was normal in the tolerized cells (Figure 1E), suggesting that tolerance at this time frame was independent of the general negative regulation of the activation/signaling pathways.

### 2.2. Prolonged Inhibition of the MEK1/2-ERKs Signaling Axis Primes RAW264.7 Cells to LPS-Induced IL-1β Expression

Since prolonged exposure, but not short-term exposure, to MEK1/2 inhibitors reversed IL-1β expression in tolerated macrophages, we further examined if MEK1/2 inhibition could enhance IL-1β expression in non-tolerated macrophages. When RAW264.7 cells were pre-treated with inhibitors targeting MEK1/2, p38, JNK, and AKT for 18 h, only the MEK1/2 inhibitors U0126 and selumetinib further enhanced IL-1β mRNA expression over the levels of non-treated cells in response to LPS (Figure 2A). Like in the tolerated cells, U0126 significantly enhanced IL-1β mRNA expression only after 18–24 h of exposure (Figure 2B) in a dose–response manner, maximally at 10 µM of U0126 (Figure 2C). In line with mRNA expression, U0126 enhanced IL-1β protein production in both tolerated and non-tolerated cells (Figure 2D). To further substantiate the role of basal MEK1/2 activity in IL-1β expression, catalytically active and inactive mutants of MEK1 were ectopically expressed for 48 h using adenoviral vectors. The expression of the catalytically active MEK1 (MEK1-CA) led to constitutive ERK activation and significantly lower IL-1β mRNA expression, whereas the expression of the catalytically inactive MEK1 (MEK1-CI) led to lower constitutive ERK activation and higher IL-1β mRNA expression than vector transfection in response to LPS (Figure 2E). Collectively, these data suggest that the constitutive MEK1/2-ERK signaling axis determines the level of IL-1β expression in LPS-activated macrophages.

### 2.3. Prolonged Inhibition of MEK1/2 Enhances the Expression of Genes Involved in Immune Response and Cytokine Production but Suppresses Genes Involved in Cell Cycle Progression and Heterochromatin Formation

To examine the overall impact of MEK1/2 inhibition and the potential mechanism of the priming effects, we performed a transcriptomic analysis in unexposed RAW264.7 cells or RAW264.7 cells exposed to U0126 (5 µM) for 18 h. U0126 induced 1190 genes and suppressed 916 genes by more than 1.5-fold (adj. *p* < 0.05; Figure 3A). To dissect the functional aspects of these gene changes, a Gene Set Enrichment Analysis (GSEA) was performed using the mouse Molecular Signature Database (M5.all.v2023). Among the mouse ontology gene sets, 678 gene sets were induced, whereas 516 gene sets were suppressed by U0126. These data sets were visualized using the Cytoscape program [32] (node cutoff: *q*-value ≤ 0.01; edge cutoff: FDR ≤ 0.5) (Figure 3B). The prominent functional clusters that were decreased by U0126 include cell cycle progression/chromosome regulation (183 nodes) and epigenetic regulation/heterochromatin formation (33 nodes), whereas the increased ones are related to immune responses/regulation (80 nodes) and cytokine/chemokine production (10 nodes) (Appendix A). Among the top 20 leading changes, the response to bacteria, IL-1 production, and the external side of the plasma membrane gene sets were positively enriched, whereas DNA replication, the mitotic cell cycle, and the chromosome organization gene sets were negatively enriched by U0126 (Figure 3C). Among the 33 epigenetic regulation/heterochromatin nodes, the 6 heterochromatin nodes contained 61 genes, which were mostly suppressed by U0126 (Figure 3D).

### 2.4. MEK1/2 Inhibition Regulates the Expression of Genes That Promote Histone H3K9 Demethylation

Among the changes induced by U0126, a decrease in genes associated with heterochromatin formation was of interest since histone modification is a key epigenetic mechanism in macrophage tolerance and training. Particularly, the H3K9-methylation-related and heterochromatin-related genes such as *Dnmt1*, *Setdb1/2* (encoding the SET domain bifurcated histone lysine methyltransferase; SETDB), *Cbx1/2/3/5* (encoding the suppressor of variegation 3–9 homolog H1 and 2; SUV39H1/2), and *Ezh2* were downregulated by U0126 (Figure 3D). To further examine the genes that are selectively involved in histone modifications, we looked into the 703 genes annotated as histone modifiers (Appendix A) and found that 30 genes were significantly changed (FC > 2; *p* < 0.05) by U0126 (Figure 4A; Appendix A). Particularly, *Cbx5* and *Suv39h1* (encoding a histone H3K9 methyltransferase) were significantly downregulated, but *Kdm7a* (encoding an H3K9 demethylase) was significantly up-regulated. A qPCR analysis also confirmed that the mRNA levels of *Suv39h1* and *Cbx5* were decreased, but *Kdm7a* was increased in the U0126-treated cells (Figure 4B). Since H3K9 and DNA methylation are correlated with gene repression [33], we examined if the inhibition of the H3K9 histone demethylases, KDM5 and 7, negates the priming effect of U0126 and if the inhibition of histone and DNA methyltransferases mimics the U0126 priming effect. As shown in Figure 4C, neither the KDM5 nor 7 inhibitors prevented the priming effect. Among the H3K9 and DNA methyltransferase inhibitors examined, the H3K9 methyltransferase G9a inhibitor BIX01294 (BIX) was able to prime cells to LPS (Figure 4D). Since the G9a-CBX5 complex is required for H3K9 methylation and gene repression [34,35,36], the BIX effect may also indicate a key role of CBX5 in the priming effect. To confirm the role of CBX5, RAW264.7 cells were stably transfected with eGFP alone (vector control) or eGFP-conjugated CBX5 (eGFP-CBX5; Figure 4E, right panel) and we examined if the ectopic expression of eGFP-CBX5 inhibited the priming. RAW264.7 cells transfected with the vector alone were primed by U0126, but not in the eGFP-CBX5-transfected cells (Figure 4E, left panel). These data suggest that, although multiple factors could have been cooperatively involved in H3K9 demethylation, CBX5 likely played a key role in the priming effect of U0126.

### 2.5. Inhibition of MEK1/2 Decreases H3K9 Trimethylation in the Genomic Region of IL-1β, CXCL10, and Tumor Necrosis Factor (TNF)α

Since H3K9 demethylation was involved in the priming of IL-1β transcription, we further examined if the H3K9 methylation levels were lowered in the IL-1β gene region when the cells were treated with U0126. To find potential H3K9me3-associated sites in the macrophages, the H3K9me3 ChIP-seq database available in the GEO database (GSE107227; peritoneal macrophages) was visualized in the UCSC mouse mm9 Genome browser. Two prominent H3K9me3 peaks close to the IL-1β promoter and intragenic regions were selected for analysis: Site 1 is located at ~7500 bp upstream of the IL-1β transcription start site (highlighted in gray), and Site 2 is located within intron 3 (highlighted in pink) (Figure 5A, left panel). H3K9me3 chromatin immunoprecipitation (ChIP)-qPCR analysis, using primers targeting Site 1 or Site 2, showed that U0126 significantly decreased the H3K9me3 levels in Site 1 (Figure 5A, right panel). Site 2 also showed a decreasing trend, but it was not statistically significant. We then further examined if U0126 could also enhance the expression of other inflammatory cytokines including IL-6, CXCL10, and TNFα. Similar to IL-1β, these inflammatory cytokines were also further induced when the cells were treated with U0126 for 18 h (Figure 5B). We then examined two H3K9me3 peaks (the one within the genomic area and the other proximal to the promoters; Figure 5C, left panel). Except in IL-6, one of the two regions in both CXCL10 and TNFα showed significantly lower H3K9me3 levels in the U0126-treated cells than in the non-treated cells (Figure 5C, right panel). These results suggest that U0126 primed not only IL-1β, but also other inflammatory cytokines, likely by decreasing the repressive H3K9me3 levels. 

### 2.6. Inhibition of the MEK1/2-ERK Signaling Axis Primes IL-1β and Other Inflammatory Cytokines in Primary Bone-Marrow-Derived Macrophages

To confirm the role of the MEK1/2-ERK signaling axis in training IL-1β and other inflammatory cytokines in primary macrophages, bone-marrow-derived macrophages (BMDMs) were prepared from C57BL/6 mice by culturing bone marrow cells in the presence of macrophage colony-stimulating factor (M-CSF). BMDMs were non-tolerized or tolerized with LPS (10 ng/mL) for 24 h in the presence or absence of U0126 or BIX and then activated by LPS (100 ng/mL) for 6 h. Similar to the RAW264.7 cells, the tolerated BMDMs failed to express IL-1β mRNA, but the BMDMs tolerized in the presence of U0126 or BIX were able to express IL-1β mRNA similarly to the non-tolerized cells (Figure 6A, left panel). Also, the BMDMs primed with U0126 or BIX expressed higher levels of IL-1β mRNA (middle panel) and protein levels (right panel). U0126 and/or BIX also reversed tolerization in the mRNA expression of IL-6 and TNFα (Figure 6B, upper panel), and further enhanced IL-6, TNFα, and CXCL10 in non-tolerized cells (Figure 6B, lower panel).

## 3. Discussion

This study demonstrated that the prolonged (>18 h) inhibition of the MEK1/2-ERK signaling axis reverses tolerance and primes macrophages in expressing IL-1β and other key inflammatory cytokines such as IL-6, TNFα, and CXCL10. The tolerance-reversing effect via MEK1/2-ERK inhibition may not implicate that the signaling axis is directly involved in the tolerance process since U0126 treatment during the first 6 h of tolerance (when the signaling axis is activated) failed to reverse tolerance (Figure 1C), and U0126 alone was able to prime non-tolerized cells (Figure 2A). Also, failure in reversing tolerance via the short-term treatments of U0126 suggests that the effect was due to a delayed adaptation, likely through an epigenetic event rather than a direct signaling event. The prolonged inhibition of the signaling axis led to prominent transcriptomic changes that enriched transcripts involved in immune responses/signaling and cytokine/chemokine production, but repressed transcripts involved in cell proliferation and lipid biosynthesis pathways (Figure 3). Also, cells transfected with the active MEK1 mutant expressed lower IL-1β expression levels, but cells with the inactive MEK1 mutant expressed higher IL-1β expression levels when compared to the vector-transfected control cells (Figure 2E). These results suggest an unexpected role of the MEK1/2-ERK signaling axis that suppresses inflammatory cytokine production at a basal condition.

The tolerance-reversing and priming effects of MEK1/2-ERK inhibition are reminiscent of the effects induced by ultra-low doses of LPS [37,38,39], BCG vaccine [40], C. albicans [41] and its cell wall component β-glucan [4,42], oxidized low-density lipoprotein (ox-LDL) [43,44], and the antineoplastic agent known as carboplatin [45]. These stimuli commonly reverse LPS tolerance and/or train/prime macrophages through epigenetic reprogramming. One of the key epigenetic mechanisms in reversing tolerance is mediated by introducing/stabilizing the transactivation histone markers, namely H3K4me2/3 and H3K27ac, or removing the silencing histone marker H3K9me2/3 [10,46,47]. These histone modifications at the promoters and enhancers are introduced in response to training stimuli in a gene-specific manner. We found that gene sets involved in heterochromatin formation were suppressed by U0126 (Figure 3D), and among them, *Kdm7a*, *Suv39h1*, and *Cbx5* pointed to H3K9 demethylation as a potential modification involved in the U0126 priming effect (Figure 4). H3K9me3 is a facultative and constitutive gene silencer rendering transcription repression, heterochromatin formation, and DNA methylation in lineage-specific cell differentiation and maintenance [48,49]. In macrophages and dendritic cells, high levels of H3K9 methylation are associated with low inflammatory cytokine production [50,51,52,53]. Interestingly, H3K9 methylation suppresses the activation of all MAPKs (ERK, p38, and JNK) and NF-κB, and suppresses various proinflammatory cytokines in the heart [54]. 

Thus, the lowering of the H3K9 methylation levels by U0126 could have been attributed to the overall enhancement of transcripts involved in immune response genes (Figure 3B). 

The dynamic methylation status of H3K9 is determined by the counteracting methyltransferases and demethylases. The known H3K9 mono-/di-methyltransferases are G9a and G9a-like protein 1 (GLP1), and the H3K9 di-/tri-methyltransferases are SUV39H1, SUV39H2, SETDB1, and SETDB2 [48,49]. These methyltransferases have redundancy in some functions, but also have unique cell type-specific roles. Among them, U0126 significantly repressed the transcription of *Suv39h1* (Figure 4A,B). Previously, a low expression of SUV39H1 was implicated in high inflammatory cytokine production in peripheral blood monocytes [55], and its overexpression in macrophages decreased inflammatory cytokines [56]. Therefore, the repression of *Suv39h1* is likely contributed to the priming effects of U0126. However, the selective SUV39H1/2 inhibitor, chaetocin, failed to mimic the priming effect of U0126 (Figure 4D). We speculate that non-specific [57] or cytotoxic effects at effective doses [58] could have complicated the results. Unlike chaetocin, the G9a inhibitor BIX01294 was able to mimic the priming effect of U0126 (Figure 4D and Figure 6) in both RAW264.7 cells and BMDMs, suggesting that the inhibition of H3K9 methylation could be associated with the priming effect. In addition to the repression of the H3K9 methyltransferase, H3K9me2/3 demethylation could have been enhanced by the induction of KDM7, which is a known H3K9 and H3K27 dual demethylase [59]. However, KDM7 inhibition alone did not affect the priming effect of U0126 in IL-1β expression (Figure 4C), suggesting its minor role.

Another histone H3K9 methylation regulator is *Cbx5* (Figure 4). CBX5 is one of the three HP1 family members, including CBX1/HP1-β, CBX3/HP1-γ, and CBX5/HP1-α, that directly bind to H3K9me2/3 and function as H3K9me2/3 readers [35,36]. These CBX members play key roles in spreading H3K9 methylation markers by recruiting the H3K9 methyltransferases SUV39H1/2 and other factors that induce chromatin compaction and gene silencing. Among the three members, the expression of *Cbx5* in RAW264.7 cells was the most abundant based on the transcript counts, with average read counts of 34454, 15425, and 7087 for *Cbx5* (gene id: 12419), *Cbx3* (gene id: 12417), and *Cbx1* (gene id: 12412), respectively. Importantly, *Cbx5* expression was repressed by U0126 (Figure 4A,B). More conclusively, RAW264.7 cells that were stably transfected with CBX5 failed to be primed by U0126 (Figure 4E). CBX5 is known to directly interact with SUV39H1 [60] or G9a [34], which are required for optimal gene repression. Based on the results shown by G9a inhibition (Figure 4E and Figure 6) and the ectopic expression of CBX5, we speculate that lowering the H3K9me3 levels by downregulating the G9a/CBX5/H3K9me2/3-mediated gene silencing pathway was a mechanism involved in the priming effect of the MEK1/2-ERK inhibition. More evidently, U0126-trained macrophages harbor lower H3K9me3 levels than those of non-treated cells in the genomic regions of IL-1β and other cytokines including CXCL10 and TNFα (Figure 5).

Previously, it was shown that the platinum-based anticancer drug, carboplatin, exerts both anti-proliferative and tolerance-reversing effects [45,61], whereas the anti-proliferative and cell death effects are due to its DNA alkylating activities that cause DNA damage [61], and the tolerance-reversing effects are likely caused by the demethylation of H3K9me3 in macrophages [45]. Similar to our results, carboplatin, when treated together with LPS during the tolerance-inducing stage, derepressed TNFα and IL-6 expression in LPS-tolerated macrophages. It was suggested that carboplatin selectively represses the expressions of *Suv39h1* and *Cbx5*, resulting in low H3K9 methylation. Since both U0126 and carboplatin can cause cell cycle arrest in macrophages, it is not clear if cell cycle arrest was the common contributor to priming macrophages through H3K9 demethylation. However, the inhibition of cell cycle progression by the chemical inhibitors AZD5438 and nocodazole failed to train the macrophages but rather inhibited IL-1β transcription (Appendix A), and MEK/12-ERK inhibition had similar tolerance-reversing and priming effects in less proliferative BMDMs as the highly proliferative RAW264.7 cells (Figure 6), ruling out the involvement of cell cycle arrest per se in training.

To date, several signaling mechanisms involved in training macrophages by β-glucan, BCG, and oxLDL have been reported. β-glucan (a cell wall component of *C. albicans*) trains and reverses LPS tolerance in the expression of proinflammatory cytokines in macrophages [3,4]. Mechanistically, β-glucan activates dectin-1 to induce the enrichment of genes that are involved in the cAMP-PKA and cortisol-mediated signaling axes that increase H3K4 methylation and H3K27 acetylation in promoters and enhancers by recruiting H3K4 methyltransferases and histone acetylases, likely through transcription factors like ATF1/7 and CRE binding protein 3 [42]. BCG- and oxLDL also induce training of proinflammatory cytokines such as TNFα and IL-1β by activating NOD2 and TLRs, respectively, leading to H3K4 methylation and H3K27 acetylation [40,43,62]. Concurrently, these training stimuli shift metabolic pathways toward high glycolytic and low oxidative phosphorylation pathways [63,64,65,66]. Preferred glycolytic metabolism is a key component of the training process through the production of critical metabolic intermediates, such as acetyl-CoA, α-ketoglutarate, fumarate, and succinate [67]. The shift of the metabolic pathway is likely mediated by activating the AKT/mTOR/HIF-1α signaling axis [66,67]. The inhibition of the MEK1/2-ERK signaling axis is also shown to activate the PI3K/AKT/mTOR pathway in tumor cells and macrophages [68,69], which could have caused changes in the metabolic processes. Consistently, the U0126-treated cells downregulated most genes involved in lipid metabolism (Appendix A), suggesting that a compensatory activation of PI3K/AKT could have also contributed to the training effects through H3K4 methylation and H3K27 acetylation. Therefore, the involvement of the PI3K/AKT/mTOR/HIF-1α signaling axis in the MEK1/2-ERK inhibition warrants further studies.

As expected, the MEK1/2-ERK signaling axis is required for the optimal expression of inflammatory cytokines in macrophages (Figure 1D) through activating various downstream signaling molecules, transcription factors, mRNA stability mechanisms, and chromatin remodelers [70,71,72]. Activated ERK translocates to the nucleus and regulates transcription via both kinase-dependent and -independent mechanisms. It is well established that ERK co-localizes with pre-existing c-FOS (an AP-1 transcription factor involved in various cytokine productions), enabling the rapid and potent transcription activation of inflammatory cytokines [73]. ERK, by activating its downstream mitogen- and stress-activated kinases (MSKs), also induces the phosphorylation of Histone 3 at Ser10 (H3S10) that displaces H3K9me2/3 [74] and recruits the transactivation remodeler SWI/SNF complex [75]. These ERK-mediated events collaboratively enhance rapid and optimal inflammatory cytokines in activated macrophages. At the same time, ERK can function as a gene repressor in IL-12 expression by activating a suppressor element [76] and interferon-γ expression by directly binding to DNA and replacing the transcription factor C/EBPβ [77]. Therefore, the inhibition of the MEK1/2-ERK signaling axis could enhance transcription by inactivating repressors or activating C/EBPβ. Also, ERK negatively regulates macrophage colony-stimulating factor (M-CSF) activation and inhibits PI3K/AKT/mTOR activation in macrophages [78]. Therefore, the prolonged inhibition of the MEK1/2-ERK signaling axis could lead to the activation of the PI3K/AKT/mTOR, which is required for the epigenetic training of macrophages. Further studies are required to unravel the involvement of these events in macrophage priming.

IL-1β plays a key role in proinflammatory responses, providing resistance to microbial infections [79,80], as well as being involved in numerous inflammatory and autoimmune diseases [81,82,83] and cancers [84]. Since macrophages are the primary source of IL-1β production, regulating the expression of IL-1β by targeting the MEK1/2-ERK signaling axis will provide a novel therapeutic tool for either promoting or suppressing IL-1β expression in macrophages.

## 4. Materials and Methods

Reagents: LPS was purchased from Invivogen; U0126 was purchased from LC laboratories, selumetinib and MK-2206 were purchased from Cedarlane (Burlington, ON, Canada); SB203580 was purchased from Selleck Chemicals (Houston, TX, USA); SP600125 and nocodazole were purchased from Calbiochem (Burlington, MA, USA); AZD-5438 was purchased from APExBIO (Boston, MA, USA), BIX01294; TC-E 5002, CPI455, and SGI-1027 were purchased from Cayman (Ann Arbor, MI, USA); and 5-azacytidine was purchased from Sigma (St. Louis, MO, USA). The pGFP-mHP1α plasmid was purchased from Addgene, Watertown, MA, USA (#181900), and eEGFP-C1 vector control was purchased from BD Biosciences Clontech, Franklin Lakes, NJ, USA (#6084-1). The pro-IL-1 β antibody (1:1000 dilution) was a kind gift from Dr. Aurigemma (NCI-Frederick Cancer Research and Development Center, Frederick, MD, USA). The primary antibodies for mouse β-actin, including p38, IκB, and eGFP, were purchased from Rockland, Bridgeport, NJ, USA (#600-401-886; 1:2000 dilution), Cell Signaling Technologies, Danvers, MA, USA (#9212, #4814; 1:2000 dilution), and Clontech Lab., Mountain View, CA, USA (#632381; 1:1000 dilution), respectively; the secondary antibodies for rabbit IgG-HRP and mouse IgG-HRP were from Santa Cruz, Dallas, TX, USA (#SC-2357) and Thermo Scientific, Waltham, MA, USA (#32430), respectively. A polyclonal anti-histone H3-K9 antibody was purchased from AbClonal, Woburn, MA, USA (#A2360).

Cell Culture and Transfection: RAW264.7 cells (purchased from ATCC) were maintained in complete DMEM (Sigma Aldrich) supplemented with 8% heat-inactivated fetal bovine serum and 100 U/mL penicillin-streptomycin. Bone marrow was isolated from the tibia and femurs of 6–8-week-old male C57BL/6 mice (Charles River Laboratory) as previously described [85]. Cells were treated with 10 ng/mL LPS to tolerize with or without various inhibitors targeting MEK1/2 (U0126 and selumetinib), p38 MAPK (SB203580), JNK (SP600125), AKT (MK-2206), NF-κB (NF-κBi), CDK1/2/9 (AZD-5438) or microtubule (nocodazole), G9a (BIX01294), SUV(3-9)VAR (Chaetocin), KDM2/7 (TC-E 5002), KDM5 (CPI455), DNMT1/3a/3b (SGI-1027), and pan-DNMT (5-azacytidine)) for 30 min during LPS activation or 18–24 h before activation. Cells were then washed with PBS and activated by 100 ng/mL LPS in fresh complete growth media for various time points. RAW264.7 cells were stably transfected with 2.5 μg of the pGFP-mHP1α or the pEGFP-C1 vector control plasmids using the Lipofectamine 3000 transfection kit following the manufacturer’s protocol. Cells cultured for 18–24 h and stably transfected cells were selected by culturing media containing G418 (2–5 mg/mL) for two weeks. Stably transfected colonies were pooled and used for the subsequent experiments. For adenoviral vector transfection, a replication-defective adenoviral vector (Ad5-CMV vector), or harboring either catalytically active mutant MEK1 (Ser 217 Glu and Ser 221 Asp; Ad5-CMV-MEKca) [86] or inactive mutant MEK1 (Ser 221 Ala; Adv5-CMV-MEK1dn; Seven Hills Bioreagents, Cincinnati, OH, USA) were prepared with approximately 1 × 10^8^ MOI/mL from cell lysate of HEK293 cells as previously described [87]. RAW264.7 cells were infected with adenovirus at about 15 MOI/cell for 24 h. Cells were then washed and cultured with fresh cell culture media for the next 48 h. Cells were then activated with LPS (100 ng/mL) for 30 min (immunoblot analysis) for examination of short-term signaling mediators or for 6 h (qPCR analysis or immunoblot analysis for cytokine expression).

Quantitative Real-Time PCR (qPCR): qPCR was carried out as previously described [88]. Briefly, the isolation of total cellular RNAs and reverse transcription were performed using TRIzolTM (Ambion) and M-MuLV reverse transcriptase (New England Biology, Ipswich, MA, USA). The qPCR was conducted using the Rotor-Gene RG3000 instrument (Montreal Biotech Inc., Dorval, QC, Canada) with the 2× Universal Sybr Green Fast qPCR Mix (AbClonal) following the manufacturer’s protocol. Data are represented as fold change to the control or LPS-treated group after normalization to the glyceraldehyde 3-phosphate dehydrogenase (GAPDH) and beta-2-microglobulin (B2M) housekeeping genes. Primers were designed using the NCBI Primer Blast program, and each primer set was validated for its amplification efficiency within 100 ± 10% (Appendix A). 

Immunoblotting: Immunoblotting was performed as previously described [69]. Briefly, cells were detached from plates with PBS containing 1 mM EDTA and lysed with an ice-cold lysis buffer containing 20 mM MOPS (pH7.2), 2 mM EGTA, 5 mM EDTA, 1 mM Na3VO4, 40 mM β-glycerophosphate, 30 mM sodium fluoride, 20 mM sodium pyrophosphate, 0.1% SDS, 1% Triton X-100, and phosphatase inhibitor tablets (Pierce Biotechnology, Waltham, MA, USA). Cell extracts were then electrophoretically resolved in 10% SDS-polyacrylamide gels, and then transferred onto 0.2 μm nitrocellulose membranes. The membranes were blocked at room temperature for 1 h with 5% (*w*/*v*) skim milk, and then incubated overnight at room temperature with primary antibodies, followed by the corresponding secondary antibody for 60 min. Membranes were exposed to the BioRad Clarity Max Western ECL system (Mississauga, ON, Canada), and images were obtained using the BioRad Chemidoc XR+ System (Quantity One 4.6.4, BETA VERSION). Densitometry analysis of the bands was performed using the ImageJ 1.46r program.

RNA Sequencing and Transcriptomic Analysis: Total RNAs were prepared using the Qiagen RNeasy kit according to the manufacturer’s protocol, and mRNAs were sequenced using Illumina NovaSeq PE100 sequencer with 50 million reads per sample in Genome Quebec. Sequenced reads were aligned with NCBI37/mm9 mouse genome. Resulting BAM files were used to count for matched genes using the FeatureCount tool (with built-in gene annotation file, using default settings) [89], and fold of changes and dispersions were estimated using the DESeq2 tool (c7 < 0.05) [90] in the Galaxy platform. The differential gene expression and gene counts were then visualized in volcano plots and bubble heat maps using GraphPad Prism 10 and R Studio, respectively. To perform Gene Set Enrichment Analysis (GSEA), the Limma-voom tool [91] was used to filter out counts lower than 10 in the Galaxy, and the resulting differential expression tables were used for GSEA [92], followed by Cytoscape visualization [32,93] with a node cut-off of 0.01 *q*-value and an edge cut-off of 0.5. Specific pathways were also further visualized using the KEGG pathways in Pathview [94] in the Galaxy platform.

ChIP-qPCR: ChIP analysis was conducted as described previously [95]. Briefly, MEK1-inhibited (5 μM U0126 for 18H) or non-treated control RAW264.7 cells were cross-linked with 1% formaldehyde, which was terminated with 125 mM glycine. Cells were then lysed and sonicated for 21 cycles of 30 s on and 30 s off at 4 °C using the Bioruptor UCD-200TM-EX water bath sonicator from Diagenode. Sonicated chromatin was incubated with H3K9me3 antibody (#A2360, AbClonal, Woburn, MA, USA) and conjugated to protein G DynaBeads overnight at 4 °C. Bound beads were washed once with low-salt wash buffer (0.1% SDS, 1% Triton X-100, 2 mM EDTA, 20 mM Tris-Cl (pH 8.0), and 150 mM NaCl), once with high-salt buffer (0.1% SDS, 1% Triton X-100, 2 mM EDTA, 20 mM Tris-HCl (pH 8.0), and 500 mM NaCl), once with LiCl buffer (0.25 M LiCl, 1% NP-40, 1% sodium deoxycholate, 1 mM EDTA, and 10 mM Tris-Cl (pH 8.1)), and twice with Tris-EDTA buffer at pH 8. Immunocomplexes were eluted with 300 μL of elution buffer (1% SDS; 0.1 M NaHCO_3_) with 200 mM NaCl overnight at 65 °C. DNA was purified using the Qiagen QIAquick Spin Columns according to the manufacturer’s protocol. RT-qPCR was then conducted using the percent input method. A complete list of the primers used and their targets can be found in Appendix A.

Enzyme-Linked Immunosorbent Assay (ELISA): Cell culture media samples were collected after culturing 60,000 cells per 96-well for 18 h. IL-1β amounts in the samples were measured using the Peprotech IL-1β Mini ABTS ELISA Development Kit according to the manufacturer’s protocol.

Statistics: Data were analyzed using Microsoft Excel (2016) and GraphPad Prism 10. The results are presented as individual replicates with at least three independent repeats. Statistical significance was determined by using a one-way ANOVA test with Dunnett’s multiple comparisons test, a two-way ANOVA with Dunnett’s multiple comparisons test, or unpaired t-tests. Statistical significance was defined as *p* < 0.05.

## 5. Conclusions

This study found that the prolonged inhibition of the MEK1/2-ERK-RSK signaling axis reverses LPS tolerance and primes macrophages in the expression of IL-1β and other inflammatory cytokines by inhibiting H3K9me2/3-mediated gene repression. Therefore, the constitutive activation of the MEK1/2-ERK signaling axis may play a key role in suppressing inflammatory responses and cytokine expression in murine macrophages.

## Figures and Tables

**Figure 1 ijms-24-14428-f001:**
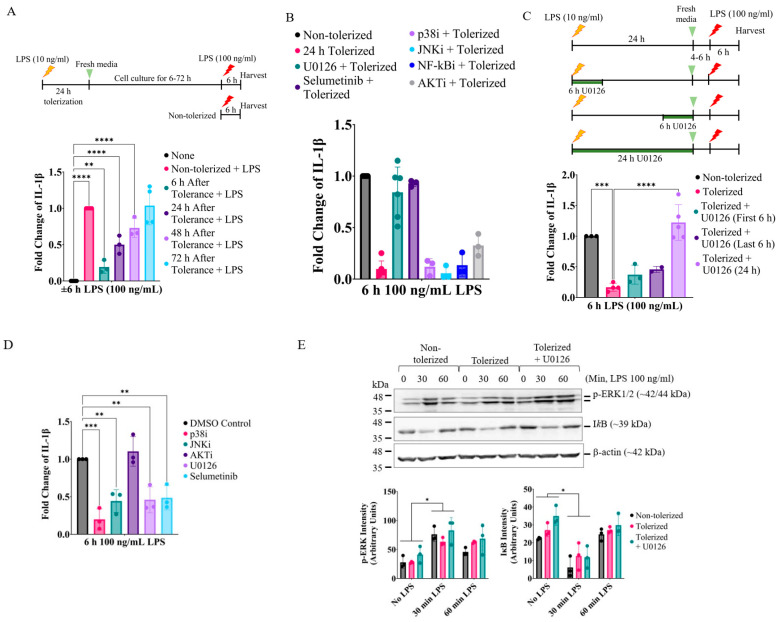
Prolonged MEK1/2 inhibition reverses LPS tolerance in IL-1β mRNA expression without affecting ERK and NF-κB activation. (**A**) RAW264.7 cells were tolerized with LPS (10 ng/mL) for 24 h and replated in fresh media for 6–72 h. Cells were then activated with LPS (100 ng/mL) for 6 h. **Upper** panel: A diagram describing the experimental procedure. **Lower** panel: Expression of IL-1β mRNA was measured via RT-qPCR. (**B**) Cells were tolerized for 24 h in the presence of signaling pathway inhibitors targeting MEK1/2 (U0126; 5 µM), p38 MAPK (p38i, SB203580; 5 µM), JNK (JNKi, SP600125; 5 µM), NF-κB (NF-κBi; 2 µM), or AKT (AKTi, MK-2206; 5 µM). Cells were then replated with fresh media for 4 h and then activated with LPS for 6 h. Expression of IL-1β mRNA was measured via RT-qPCR. (**C**) The MEK1/2 inhibitor U0126 was added at the first 6 h, the last 6 h, or throughout the 24 h tolerization period. **Upper** panel: A diagram describing the experimental procedure. **Lower** panel: Expression of IL-1β mRNA was measured via RT-qPCR. (**D**) Key signaling inhibitors were added 30 min before activation and activated with LPS (100 ng/mL) for 6 h. Expression of IL-1β mRNA was measured via RT-qPCR. (**E**) Cells were non-tolerized, tolerized, and tolerized in the presence of U0126 (5 µM) for 24 h. Cells were then replated with fresh culture media for 4–6 h and then activated by LPS (100 ng/mL) for 0, 30, or 60 min. **Upper** panel: Activation of ERK1/2 and degradation of inhibitor (I)κB were examined via immunoblotting against phospho(p)-ERK1/2, inhibitor (I)κB, and β-actin (loading control). **Lower** panel: Relative amounts of p-ERK and IκB over β-actin (loading control) were expressed. (**A**–**D**) A one-way ANOVA test followed by Dunnett’s multiple comparisons test was performed for statistical analysis (*n* = 3; ** *p* ≤ 0.01, *** *p* ≤ 0.001, **** *p* ≤ 0.0001). (**E**) Student’s *t*-test was performed between no LPS vs. 30 min LPS treatment groups in non-tolerized, tolerized, and tolerized + U0126 groups (*n* = 3, * *p* ≤ 0.05).

**Figure 2 ijms-24-14428-f002:**
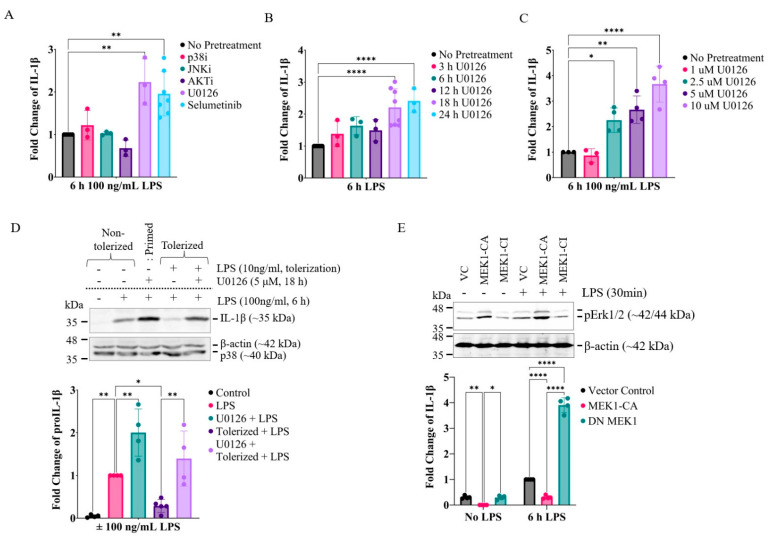
Inhibition of the MEK1/2-ERK signaling pathway enhances IL-1β expression in non-tolerized RAW264.7 cells. (**A**–**C**) Cells were treated with key signaling inhibitors for 18 h (**A**), for varying durations (3–24 h) with U0126 (**B**), or varying concentrations of U0126 for 24 h (**C**). Cells were then activated with LPS (100 ng/mL) for 6 h, and the expression of IL-1β mRNA was measured via qPCR. (**D**) Production of IL-1β was measured via immunoblotting. Cells were primed with U0126 or tolerized with LPS (10 ng/mL) ± U0126 for 18 h. Cells were replated with fresh media for 4 h and then activated by LPS (100 ng/mL) for 6 h. Whole-cell lysates were then used for immunoblotting against murine pro-IL-1β (**upper** panel). For loading controls, both β-actin and p38 immunoreactivities were examined. Relative amounts of IL-1β over β-actin (loading control) were expressed using LPS-activated samples as reference point 1 (**lower** panel). (**E**) Cells were infected with the Ad5-CMV vector alone (VC), the vector containing catalytically active MEK1 (MEK1-CA), or the vector containing catalytically inactive MEK1 (MEK1-CI) for 24 h. Cells were then washed and further cultured with fresh media for another 48 h. **Upper** panel: Cells were activated with LPS (100 ng/mL) for 30 min for immunoblotting against p-ERK and β-actin. **Lower** panel: Cells were activated with LPS (100 ng/mL) for 6 h, and the expression of IL-1β mRNA was measured via qPCR. A one-way ANOVA test followed by Dunnett’s multiple comparisons test was performed for statistical analysis (*n* = 3; * *p* ≤ 0.05, ** *p* ≤ 0.01, **** *p* ≤ 0.0001).

**Figure 3 ijms-24-14428-f003:**
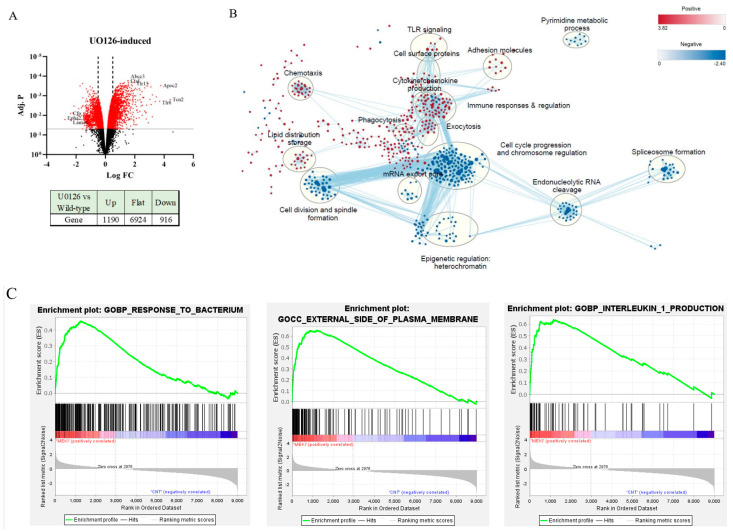
Transcriptomic analysis of RAW264.7 cells primed with U0126. (**A**) Total mRNAs of non-treated and U0126-treated cells (5 µM, 18 h) were sequenced via Illumina sequencing, and differential gene expression was analyzed using the FeatureCount and Limma-voom tools in the Galaxy platform as described in Section 4. Results shown in the volcano plot indicate the top 3 genes in the fold of increase, decrease, and adjusted *p* value (Adj. P) categories. Red dots indicate genes with Adj. *p* < 0.05; dotted vertical lines indicate fold of change (FC) > 1.5. (**B**). Gene ontology analysis was performed using the GSEA program and visualized using Cytoscape. Dots represent gene sets (nodes), and dots with a common function are clustered with a circle. Red and blue dots represent positively and negatively enriched gene sets, respectively. (**C**). Three positively and negatively over-represented gene ontology pathways are presented via enrichment plots, with size corresponding to gene count and color contrast corresponding to Adj. *p*. (**D**). The heat map shows the genes in the heterochromatin gene subsets. Expression values are represented as colors, where the range of colors (red, pink, light blue, and dark blue) shows the range of expression values (high, moderate, low, and lowest).

**Figure 4 ijms-24-14428-f004:**
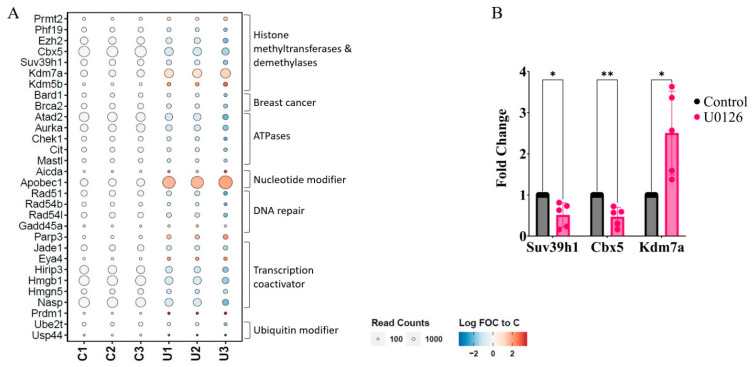
Priming RAW264.7 cells with U0126 is partly mediated by repressing genes involved in H3K9 methylation. (**A**) Based on the transcriptomic analysis, the expression of epigenetic-related genes with changes greater than 2-fold in U0126-primed cells are grouped and plotted with bubble heat maps. (**B**) Expressions of *Cbx5*, *Suv39h1*, and *Kdm7a* were confirmed via qPCR in U0126-primed and non-primed cells. Statistical significance was determined using multiple unpaired two-tailed t-tests (*n* = 3). (**C**,**D**) Cells were treated with various inhibitors for 18 h and then activated by LPS (100 ng/mL) for 6 h. Expression of IL-1β was quantified via qPCR. (**C**). Cells were treated with U0126 together with or without KDM2/7 and/or KDM5 inhibitors. (**D**) Cells were treated with inhibitors against the H3K9 methyltransferases G9a (BIX01294, 1.5 µM) and/or SUV(VAR)3–9 (chaetocin, 100 nM) or the DNMT inhibitors 5-azacytidine (2 µM) and SGI-1027 (10 µM). (**E**) Cells were stably transfected with the eGFP vector control (Vector) or the GFP-mCBX5 plasmid (CBX5). **Left** panel: Stable transfections were confirmed via eGFP or eGFP-CBX5 Western blots, using p38 antibody as a loading control. **Right** panel: Cells were activated with LPS (100 ng/mL) for 6 h, and IL-1β expression was measured with qPCR. Statistical significance was determined using one-way ANOVA tests followed by Dunnett’s multiple comparisons test (*n* = 3; * *p* ≤ 0.05, ** *p* ≤ 0.01, *** *p* ≤ 0.001).

**Figure 5 ijms-24-14428-f005:**
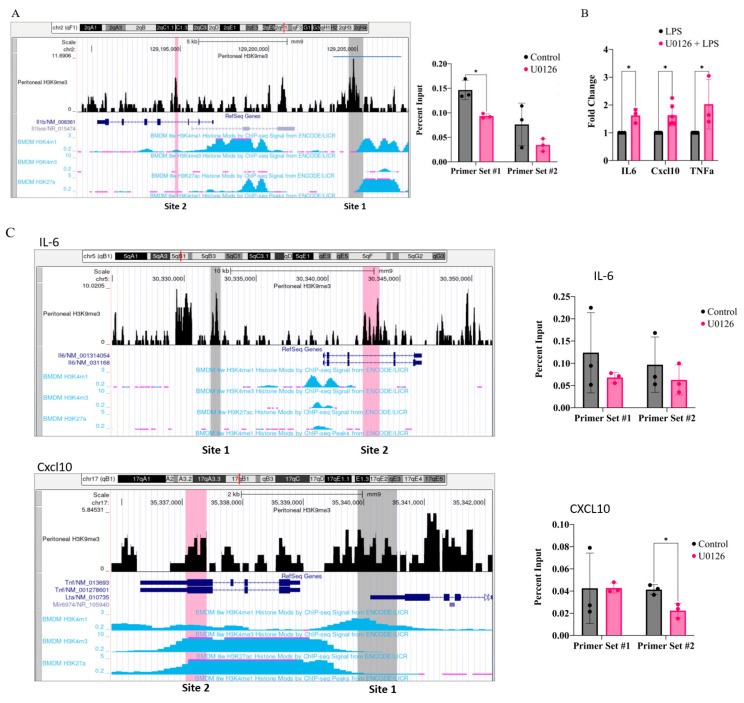
Inhibition of the MEK1/2-ERK signaling pathway decreases H3K9me3 levels proximal to IL-1β, CXCL10, and TNFα gene areas. (**A,C**) Left panel: H3K9me3 peaks in murine peritoneal macrophages from the GEO database (GSE107227), together with the built-in BMDM H3K4me1/3 and H3K27ac from the ENCODE database, were visualized in the UCSC mouse genome browser (mm9). For each gene, two peaks were selected: one was located upstream of the promoter (Site 1), and one was in an intragenic region (Site 2). (**A**) Right panel: RAW264.7 cells were treated with or without U0126 for 18 h, and associations of H3K9me3 with the IL-1β genomic area were examined via ChIP-qPCR as described in Section 4. (**B**) RAW264.7 cells were treated with or without U0126 for 18 h before activation with LPS (100 ng/mL) for 6 h. Expressions of IL-6, TNFα, and CXCL10 were analyzed via qPCR. (**C**) Right panel: Similar to (**A**), associations of H3K9me3 with the IL-6, TNFα, and CXCL10 genomic areas were examined via ChIP-qPCR. Statistical significance was determined using multiple unpaired two-tailed *t*-tests (*n* = 3; * *p* ≤ 0.05).

**Figure 6 ijms-24-14428-f006:**
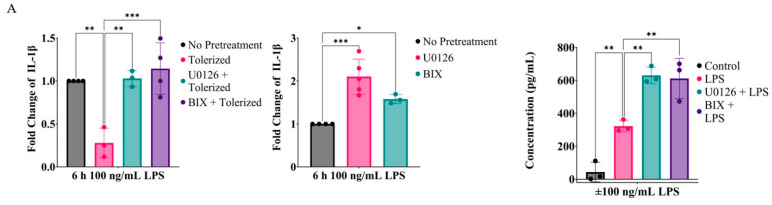
Inhibition of the MEK1/2-ERK signaling pathway reverses LPS tolerance and enhances inflammatory cytokine expression in BMDMs in a G9a-dependent manner. (**A**) BMDMs were tolerized with LPS for 24 h together with or without U0126 or BIX01294 (**left** panel) or treated 24 h before activation (**middle** panel). Cells were then activated with LPS (100 ng/mL) for 6 h for quantification of IL-1β mRNA or for 18 h for IL-1β protein levels (**right** panel). (**B**) Similarly, expressions of IL-6, TNFα, and CXCL10 mRNAs in BMDMs activated by LPS tolerized with or without U0126 or BIX, and non-tolerized cells pretreated with U0126 or BIX were analyzed via qPCR. One-way ANOVA tests with Dunnett’s multiple comparisons test were conducted to calculate significance (*n* = 3; * *p* ≤ 0.05, ** *p* ≤ 0.01, *** *p* ≤ 0.001, **** *p* ≤ 0.0001).

## Data Availability

RNA-sequencing data on wild-type and U0126-primed RAW264.7 cells will be available from the GEO database once this manuscript is accepted.

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
