# Peer review of "Prolonged Inhibition of the MEK1/2-ERK Signaling Axis Primes Interleukin-1 Beta Expression through Histone 3 Lysine 9 Demethylation in Murine Macrophages"

_ijms, 2023, doi:10.3390/ijms241914428_

Round 1

Reviewer 1 Report

Authors must check and proofread and correct missing symbols and numbers in several places.  figures 3, 5 and portions of the supplemental are too small to read.  Even at 400% mag I cannot discern what is being presented. Legends need correction.  Data should be larger easier to read and digest the data.

English is fine except for places where symbols are missing, caps is missing and hyphens are not consitent.

Author Response

Thank you for the positive and constructive comments on the manuscript submitted. The revised version of the manuscript addressed the concerns and suggestions.

  • Larger and higher resolution images in Figs 3 and 5 are now provided.
  • For Supplementary tables, only one merged PDF file is allowed to be uploaded. We will talk to the administrator to upload actual Excel files if possible.  
  • Legends and text missing symbols and capitalization of text are corrected.

Reviewer 2 Report

The study “Prolonged inhibition of the MEK1/2-ERK signaling axis primes IL-1β expression through H3K9 demethylation in murine macrophages” by Rachel Low et al., is an interesting one, in which the authors have characterised the mechanism of reversing tolerance in activated macrophages. Macrophages respond very dynamically to the stimulations from their surrounding in the body and the outcomes determine whether they have developed tolerance or are primed to induce inflammation. The authors by using macrophage cell line RAW264.7 and also primary bone marrow-derived macrophages show that inhibition of MEK1/2-ERK signalling, depending on the length of inhibition can have interesting outcome on determining the function of macrophages. Whereas, short duration inhibition (6h) resulted in the macrophages becoming tolerant, prolonged inhibition (18h) of MEK1/2-ERK signalling led to the macrophages becoming active and inducing inflammation. They have measured the inflammatory cytokine IL-1b expression at RNA and protein levels and by adopting several approaches, clearly show that prolonged inhibition of MEK1/2-ERK signalling leads to increased IL-1b production upon high dose LPS stimulation. This increase in inflammatory cytokine production was due to significant epigenetic changes at the IL-1b regulatory regions. According to the authors, it is the decrease in histone H3K9 methylation that play an essential role in the inflammatory cytokine gene expression due to inhibition of MEK1/2-ERK signalling.

Comments

It is a well worked out and well written study. The methods used in the study are appropriate. The authors need to discuss the point as to why short duration inhibition of MEK1/2-ERK signalling failed to reverse tolerance, whereas 18h inhibition could. What are the underlying mechanisms that were missing after short duration stimulation?

In the first paragraph of the results section it should be Fig. 1C instead of Fig. C.  

Author Response

Thank you for the positive and constructive comments on the manuscript submitted. The revised version of the manuscript addressed the concerns and suggestions.

It is a well worked out and well written study. The methods used in the study are appropriate. The authors need to discuss the point as to why short duration inhibition of MEK1/2-ERK signalling failed to reverse tolerance, whereas 18h inhibition could. What are the underlying mechanisms that were missing after short duration stimulation?

  • This study is to examine the mechanism of the key point. We believe that the short-term inhibition of the MEK1/2-ERK does not induce an epigenetic-mediated adaptation state. On page 10, we clearly address the point. It now reads “Also, failure in reversing tolerance by the short-term treatments of U0126 suggests that the effect was due to a delayed adaptation, likely an epigenetic event, rather than a direct signaling event.”

 In the first paragraph of the results section, it should be Fig. 1C instead of Fig. C. 

  • Corrected.

Reviewer 3 Report

The manuscript entitled “Prolonged inhibition of the MEK1/2-ERK signaling axis primes IL-1β expression through H3K9 demethylation in murine macrophages by Rachel Low  et al., highlights the  extended inhibition of the MEK1/2-ERK pathway leads to a significant increase in expression of IL-1β and decreased H3K9 methylation levels in murine macrophages. The manuscript can be accepted after minor corrections.

Please see the comments below.

1. Based on the current observation, please comment on the effect of  IL-1 beta-deficient vs normal in a KO cell line or mice model.

2. A model diagram will be helpful for easy understanding of Figure 1 and the associated experiment.

3. Please mention the sample details for each lane in the Western blot.  Supporting data Figure 2D, 2E and 4E etc. Also, include the Molecular weight for the specified protein in each blot.

4. In the experimental section please mention the SDS-PAGE gel composition, the pore size of the nitrocellulose membrane, and the dilutions of all the antibodies

5. Figure 1, on y-axis please change Fold change to fold change of  IL-1β.

6. Figure S1 should be moved to the main manuscript and relative intensities should be plotted and compared.

7. Figure 1C legends on the right side is difficult to understand, adjust the legend with respect to colored dots.

8. Correct Fig C to Figure 1C.

9. Western blot for selumetinib +tolerance in pErk1/2, IkB should be compared with U1026

10. How long inhibitors targeting MEK1/2 (U0126, 5 µM), p38 MAPK (p38i, SB203580; 5 µM), JNK (JNKi, SP600125; 5 µM), NF-κB (NF-κBi, 2 µM) or AKT (AKTi, MK-2206; 5 µM) were treated ? and at what stage.

11. Figure 1b, please mention the Tolarized time point in the figure itslef.

12. Figure 1E, western blot or  IF data on expression of IL-1β will be helpful to support the current observation concerning different inhibitors.  

13. selumetinib +tolerance can be included in Figure 1b itself and Figure 1c can be moved to supporting data.

14. Figure 3 and Figure 5 are pixelated and the font is difficult to read.  Kindly provide the image with a better resolution.

15. In few experiments, LPS treatment is 30 min and in a few participants, LPS treatment is for 6 hours why?

Minor changes in the draft

Author Response

Thank you for the positive and constructive comments on the manuscript submitted. The revised version of the manuscript addressed the concerns and suggestions.

  1. Based on the current observation, please comment on the effect of  IL-1 beta-deficient vs normal in a KO cell line or mice model.
  • Page 2: IL-1β (derived from the proteolytic processing of proIL-1β gene product) is a potent inflammatory cytokine and is rapidly induced by TLR4 activation.
  • The roles of IL-1b in knockout mice and disease models are now elaborated in the Discussion section. It reads: “IL-1β plays a key role in pro-inflammatory responses, providing resistance to microbial infections [79,80], as well as being involved in numerous inflammatory and autoimmune diseases [81-83], and cancers [84]. Since macrophages are the primary source of IL-1β production, regulating the expression of IL-1β through targeting the MEK1/2-ERK signaling axis will provide a novel therapeutic tool for either promoting or suppressing IL-1β expression in macrophages.”

  1. A model diagram will be helpful for easy understanding of Figure 1 and the associated experiment.
  • Diagrams depicting the experimental procedures are added to Fig. 1A and 1C.
  1. Please mention the sample details for each lane in the Western blot.  Supporting data Figure 2D, 2E, and 4E etc. Also, include the Molecular weight for the specified protein in each blot.
  • Details of the western blots were provided in the figure labels and legends.

  1. In the experimental section please mention the SDS-PAGE gel composition, the pore size of the nitrocellulose membrane, and the dilutions of all the antibodies
  • All details are added to the text.

  1. Figure 1, on y-axis please change Fold change to fold change of  IL-1β.
  • Y-axis labels are corrected in Fig 1, as well as other figures for consistency.

  1. Figure S1 should be moved to the main manuscript and relative intensities should be plotted and compared.
  • Figure S1 is now shown in Fig. 1E with densitometry analysis graphs.

  1. Figure 1C legends on the right side are difficult to understand, adjust the legend with respect to colored dots.
  • 1C is now removed.

  1. Correct Fig C to Figure 1C.
  • Corrected.

  1. Western blot for selumetinib +tolerance in pErk1/2, IkB should be compared with U1026
  • In this study, we showed that U0126 and selumetinib had similar tolerance reversing (Fig. 1C) and priming effects in expressing IL-1β (Fig. 2A). Furthermore, we showed similar data in catalytically active and inactive MEK1-transfected cells (Fig. 2E). Since selumetinib is a well-established and therapeutically used MEK1/2-specific inhibitor, we additionally used to validate the role of the MEK1/2-ERK signaling cascades in tolerance and priming of macrophages. It is not meant to compare the two different inhibitors in reversing tolerance and priming. To address these aspects, extensive dose-response and time-response experiments for each inhibitor will likely be required, which are not within the scope of this study.

  1. How long inhibitors targeting MEK1/2 (U0126, 5 µM), p38 MAPK (p38i, SB203580; 5 µM), JNK (JNKi, SP600125; 5 µM), NF-κB (NF-κBi, 2 µM) or AKT (AKTi, MK-2206; 5 µM) were treated ? and at what stage.
  • These inhibitors were treated for 24 h at the same time when inducing LPS tolerance. The revised figure legend provides the details on the treatments.

  1. Figure 1b, please mention the Tolarized time point in the figure itslef.
  • The tolerized time is now provided in the figure legend.

  1. Figure 1E, western blot or  IF data on expression of IL-1β will be helpful to support the current observation concerning different inhibitors.  
  • 1E, now Fig. 1D, was performed to show that, unlike the priming effects of prolonged U0126 treatments, U0126 can inhibit IL-1β mRNA expression when treated during the time of activation. This data, together with Fig. 1A-C, clearly shows that the U0126 we used inhibited IL-1β mRNA expression when treated during an activation stage. This study is not meant to address the roles of different signaling cascades in expressing L-1β mRNA/protein during LPS activation. Adding protein expression data may not further support the current observation. To address these aspects, extensive dose-response and time-response experiments for each inhibitor will likely be required, which are not within the scope of this study.

  1. selumetinib +tolerance can be included in Figure 1b itself and Figure 1c can be moved to supporting data.
  • As suggested, selumetinib data were included in Fig. 1B. Since the data was included and Fig.1C is no longer needed, the data is now removed.

  1. Figure 3 and Figure 5 are pixelated and the font is difficult to read.  Kindly provide the image with a better resolution.
  • High-resolution images are now provided for Fig. 3 and 5.

  1. In few experiments, LPS treatment is 30 min and in a few participants, LPS treatment is for 6 hours why?
  • For immunoblots examining signaling activation, short-term 30-minute samples were collected. For qPCR experiments examining transcriptional activation, 6 h samples were collected.